# ESAT-6 undergoes self-association at phagosomal pH and an ESAT-6-specific nanobody restricts *M. tuberculosis* growth in macrophages

Timothy A Bates[1], Mila Trank-Greene[1], Xammy Huu Wrynla[1], Aidan Anastas[1], Sintayehu K Gurmessa[1], Ilaria R Merutka[1], Shandee D Dixon[1], Anthony Shumate[2], Abigail R Groncki[1], Matthew AH Parson[3], Jessica R Ingram[4†], Eric Barklis[1], John E Burke[3,5], Ujwal Shinde[2], Hidde L Ploegh[4], Fikadu G Tafesse[1]*

[1]Department of Molecular Microbiology and Immunology, Oregon Health & Sciences University, Portland, United States; [2]Department of Chemical Physiology and Biochemistry, Oregon Health & Science University, Portland, United States; [3]Department of Biochemistry and Microbiology, University of Victoria, Victoria, Canada; [4]Program in Cellular and Molecular Medicine, Boston Children's Hospital, Harvard Medical School, Boston, United States; [5]Department of Biochemistry and Molecular Biology, The University of British Columbia, Vancouver, Canada

**\*For correspondence:** tafesse@ohsu.edu

†Deceased

## eLife assessment

This **useful** study investigates two secreted Mycobacterium tuberculosis proteins, ESAT-6 and CFP10, using biochemical assays, including a Biolayer Interferometry assay. **Solid** experimental evidence demonstrates that ESAT-6 forms a tight interaction with CFP10 as a heterodimer at neutral pH and that ESAT-6 also forms a homodimer at acidic pH. Additional, more definitive evidence is required to describe how these proteins disrupt the phagosomal membrane. While improved compared to a previous version, the revised manuscript did not address these concerns adequately.

**Abstract** *Mycobacterium tuberculosis* (Mtb) is known to survive within macrophages by compromising the integrity of the phagosomal compartment in which it resides. This activity primarily relies on the ESX-1 secretion system, predominantly involving the protein duo ESAT-6 and CFP-10. CFP-10 likely acts as a chaperone, while ESAT-6 likely disrupts phagosomal membrane stability via a largely unknown mechanism. we employ a series of biochemical analyses, protein modeling techniques, and a novel ESAT-6-specific nanobody to gain insight into the ESAT-6's mode of action. First, we measure the binding kinetics of the tight 1:1 complex formed by ESAT-6 and CFP-10 at neutral pH. Subsequently, we demonstrate a rapid self-association of ESAT-6 into large complexes under acidic conditions, leading to the identification of a stable tetrameric ESAT-6 species. Using molecular dynamics simulations, we pinpoint the most probable interaction interface. Furthermore, we show that cytoplasmic expression of an anti-ESAT-6 nanobody blocks Mtb replication, thereby underlining the pivotal role of ESAT-6 in intracellular survival. Together, these data suggest that ESAT-6 acts by a pH-dependent mechanism to establish two-way communication between the cytoplasm and the Mtb-containing phagosome.

## Introduction

Tuberculosis (TB), caused by *Mycobacterium tuberculosis* (Mtb), is a major global health concern, claiming over 1.6 million lives annually. Mtb is highly adept at evading the immune system, and it can persist within an infected person for decades. This situation is worsened by the nearly 1.7 billion people globally who harbor a dormant Mtb infection (*Houben and Dodd, 2016*), coupled with the rising trend of drug-resistant TB cases (*World Health Organization, 2022*).

To successfully establish an infection in the host, Mtb utilizes a plethora of virulence factors delivered via its several ESX-secretion systems. Studies have indicated that Mtb has the ability to damage the phagosomal membrane, a process that is heavily reliant on the activity of the ESX-1 secretion system (*Conrad et al., 2017*; *López-Jiménez et al., 2018*; *Schnettger et al., 2017*). Currently, the most probable candidate driving ESX-1 dependent lytic activity is the well-known T cell antigen, the 6 kDa early secreted antigenic target (ESAT-6, or EsxA) (*Andersen et al., 1995*; *Sørensen et al., 1995*), after which the ESX family of T7SSs was originally named (*Gey van Pittius et al., 2001*). Secretion of the various ESX-1 substrates are known to be highly complex and interdependent, and while the full scope of ESX-1 components and substrates is still being worked out, many individual ESX-1 component knockouts are known to phenocopy the ESAT-6 deletion, indicating that ESAT-6 itself is responsible for the lion's share of the activity (*Chen et al., 2012*; *Clemmensen et al., 2017*; *Cronin et al., 2022*; *Lienard et al., 2020*; *Sanchez et al., 2020*).

Over the last few decades, there have been numerous studies focused on uncovering the possible mechanism of membrane lysis by ESAT-6 (*Augenstreich et al., 2020*; *Conrad et al., 2017*; *de Jonge et al., 2007*; *Kinhikar et al., 2010*; *Koo et al., 2008*; *Lienard et al., 2020*; *Osman et al., 2022*), and how this might lead to escape of Mtb from the phagosome (*Houben et al., 2012*). However, the field has been plagued by setbacks such as the discovery that many studies prior to 2017 showing membrane disruption by ESAT-6 were confounded by the contamination with the detergent ASB-14, which was originally added to remove endotoxin (*Conrad et al., 2017*). Consequently, most recent studies have focused on the use of intact bacteria in assays, frequently with the closely related *Mycobacterium marinum* (Mm), to avoid using recombinant proteins. The conclusions of these experiments are complicated by the interdependent nature of ESX-1 structural components, chaperones, and substrates (*Augenstreich et al., 2020*; *Osman et al., 2022*; *Santucci et al., 2022*). Mm has been a valuable model for working out ESX-1 and ESAT-6 function, and many of the key studies in the field have used Mm exclusively. However, the unique characteristics of Mm compared to Mtb can sometimes leads to differing results, and conclusions drawn from one species do not always apply to the other (*Bosserman et al., 2019*; *Carlsson et al., 2009*; *Lienard et al., 2020*; *Osman et al., 2022*; *Smith et al., 2008*). This is important because the membranolytic activity of the ESX-1 seems to be highly context dependent, and more biochemical work is needed to work out the specific contributions of ESAT-6 to the membrane disruption phenotype.

Despite extensive research, the precise mechanism of ESAT-6 action remains elusive. This study aims to bridge this gap by performing comprehensive biochemical and cellular investigations of ESAT-6 under neutral and acidic conditions. We use biolayer interferometry (BLI) to measure the tight association between ESAT-6 and CFP-10, and the pH-dependent ESAT-6 self-association. We use fluorescence microscopy to directly observe formation of large ESAT-6 complexes, and we measure the stoichiometry of ESAT-6 under different pH conditions using multi-angle light scattering. We then use molecular dynamics-based modeling to evaluate the most likely modes of ESAT-6 interaction. We also generated a novel ESAT-6-binding alpaca-derived nanobody, E11rv, to further define the significance of ESAT-6 during Mtb infection. We perform biochemical characterization of E11rv, as well as functional testing, which showed inhibition of Mtb growth inside macrophages treated with E11rv or expressing cytoplasmic E11rv.

## Results

### Recombinant ESAT-6 and CFP-10 purification

In seeking to perform functional studies of ESAT-6 and CFP-10, we produced recombinant protein using *E. coli* expression plasmids available from BEI Resources: pMRLB.7 (NR-50170) containing ESAT-6, and pMRLB.46 (NR-13297) containing CFP-10. Both of these plasmids have been used extensively in literature, including many of the most influential biochemical characterization studies (*Augenstreich*

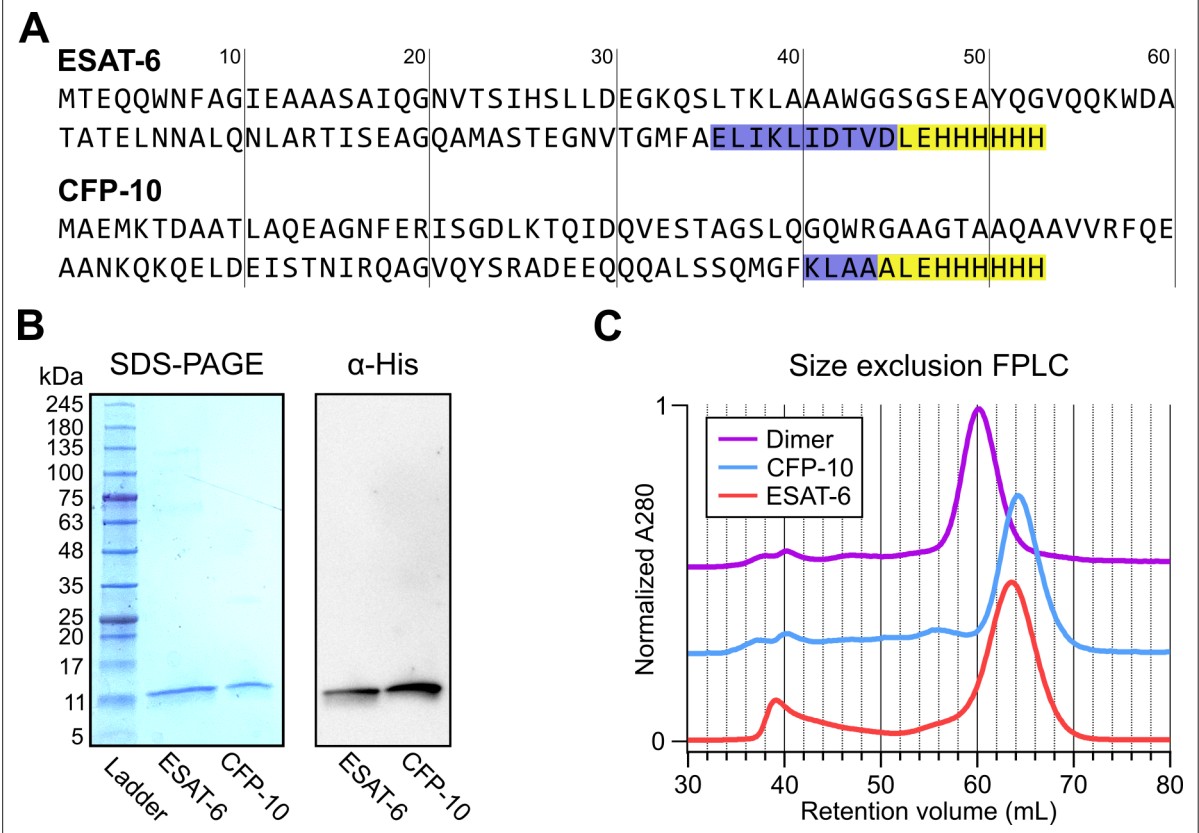

**Figure 1.** Expression of recombinant ESAT-6 and CFP-10. (**A**) Amino acid sequences derived from plasmid sequencing of pMRLB.7 (ESAT-6) and pMRLB.46 (CFP-10) plasmids available from BEI Resources. The yellow highlights indicate the expected FALE-Histidine tag amino acid sequence while the blue highlights indicate unexpected additional amino acids. (**B**) SDS-PAGE gel of purified ESAT-6 and CFP-10 with a paired anti-His$_6$ western blot. (**C**) Size exclusion chromatography traces showing the retention volumes of purified ESAT-6, CFP-10 and an equimolar mixture of both (Dimer).

The online version of this article includes the following source data and figure supplement(s) for figure 1:

**Source data 1.** SDS-PAGE and western blot.

**Figure supplement 1.** Sequencing.

**Figure supplement 2.** Whole protein mass spectrometry (MS) of recombinant ESAT-6 and CFP-10.

et al., 2020; Behura et al., 2019; Chen et al., 2019; Choi et al., 2010; Conrad et al., 2017; de Jonge et al., 2007; Eitson et al., 2012; Gallegos et al., 2008; Harrison et al., 2014; Koyuncu et al., 2021; Lim et al., 2016; Mukherjee et al., 2007; Niazi et al., 2015; Osman et al., 2022; Pathak et al., 2007; Peng et al., 2011; Refai et al., 2015; Smith et al., 2008; Stavri et al., 2012; Vesosky et al., 2010; Woolhiser et al., 2007; Xiao et al., 2016; Yang et al., 2008). The specification sheet for pMRLB.7 indicates that it includes a C-terminal FALE solubility tag and subsequent Histidine affinity tag while pMRLB.46 indicates that it contains only a C-terminal Histidine tag. We performed sequencing of both plasmids (*Figure 1A*), and noticed a stretch of 10 additional amino acids ahead of the reported tags in the pMRLB.7 plasmid (*Figure 1—figure supplement 1*). The precise amino acid sequence of the pMRLB.46 plasmid has not been reported, but it appears to contain 4 additional amino acids ahead of the expected tag. Importantly, we were able to confirm that the full and complete sequences of ESAT-6 and CFP-10 are each contained in their respective plasmids, and the inconsistencies fall within the C-terminal tag regions of each protein (*Figure 1A*).

Expression and purification by Nickel-NTA chromatography resulted in bands of the expected sizes by SDS-PAGE and His$_6$-specific western blot (*Figure 1B*). Amino acid sequences of the recombinant proteins were also confirmed by mass spectrometry (MS), where it was observed that CFP-10 undergoes N-terminal methionine excision in our *E. coli* expression system (*Figure 1—figure supplement 2*; *Giglione et al., 2004*; *Hirel et al., 1989*). Size exclusion chromatography (SEC) showed appropriate

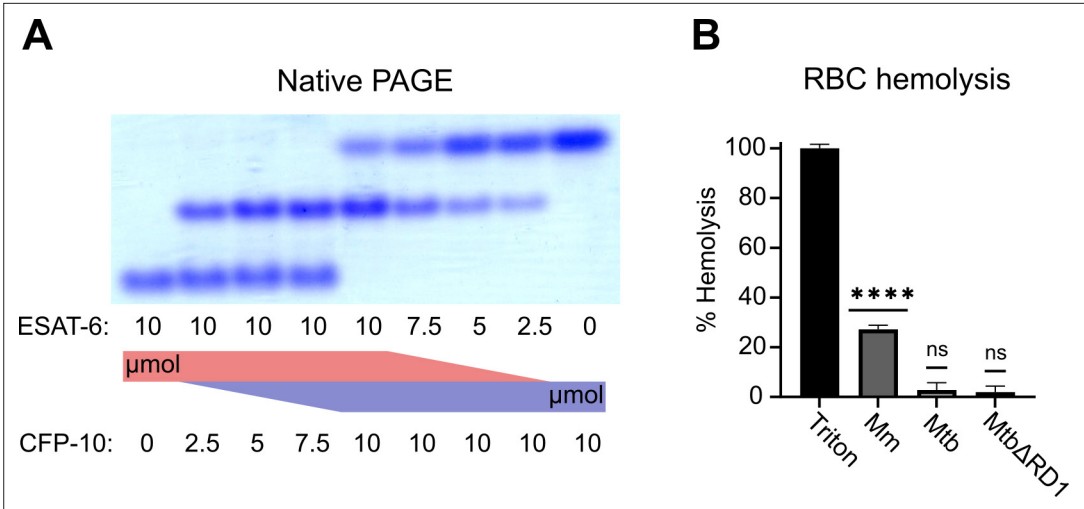

**Figure 2.** Functional testing of ESAT-6 and CFP-10. (**A**) Native-PAGE analysis of ESAT-6 and CFP-10 showing 1:1 complex formation at neutral pH. The displayed gel is representative of 3 replicates. (**B**) Contact-dependent red blood cell (RBC) hemolysis by whole bacteria, either *Mycobacterium marinum* (Mm), *Mycobacterium tuberculosis* (Mtb), or Mtb with the region of difference 1 deletion, incapable of producing ESAT-6 or CFP-10 (MtbΔRD1). Data in B represents 6 replicates. Error bars represent the 95% confidence interval. Statistical significance was determined using a one sample t test against a theoretical mean of 0. *=0.05, **=0.01, ***=0.001, ****=0.0001.

The online version of this article includes the following source data and figure supplement(s) for figure 2:

**Source data 1.** Native PAGE.

**Figure supplement 1.** Hemolysis by recombinant ESAT-6.

relative retention volumes and combining equimolar quantities of ESAT-6 with CFP-10 resulted in a leftward shift, suggesting correct dimerization (*Figure 1C*).

## Dimerization of ESAT-6 and CFP-10 at neutral pH

To validate that ESAT-6 and CFP-10 are forming a native dimer, we next performed native-PAGE with different ratios of the two proteins (*Figure 2A*). We incubated ESAT-6 and CFP-10 for 1 hr at 4 °C prior to running the gels, combining them in different ratios. The isoelectric point of ESAT-6 and CFP-10 are both quite low, for our constructs they are 5.18 (net charge –8.22 at pH 7.4) and 5.72 (net charge –5.27 at pH 7.4), respectively. Thus, we oriented the positive electrode toward the bottom of the gel. As expected, we observed that ESAT-6 ran faster than CFP-10. Combinations of each showed a distinct third band of intermediate charge which was greatest in the 1:1 mixture and absent in each pure protein, indicating formation of a 1:1 complex with no intermediate species. This aligns with our results from *Figure 1C*, and aligns with historical research on the interaction between ESAT-6 and CFP-10 (*Poulsen et al., 2014*; *Renshaw et al., 2005*; *Renshaw et al., 2002*).

Another commonly used test of ESAT-6 function is hemolysis, or the ability to damage red blood cell (RBC) membranes and release hemoglobin. Historical studies reported that ESAT-6 was able to cause hemolysis, but *Conrad et al., 2017* discovered that most of this hemolytic activity was due to detergent contamination with ASB-14, which is commonly added during the wash steps of Nickel-NTA purification to remove endotoxin. Our recombinant ESAT-6 was produced without detergent, and we confirmed that it was not able to induce hemolysis at any pH between 4.5 and 7.5 (*Figure 2—figure supplement 1*). However, the hemolysis assay still finds contemporary use when testing the ability of whole bacteria to induce membrane damage in an ESX-1 and contact-dependent manner (*Augenstreich et al., 2020*; *Bosserman et al., 2019*; *Conrad et al., 2017*; *Osman et al., 2022*). In such experiments, RBCs are mixed with log-phase bacterial cultures at high MOI and pelleted by centrifugation to force direct contact. Similar to previous reports, we found that *Mycobacterium marinum* (Mm) is capable of efficiently causing hemolysis within 2 hours (*Figure 2B*). In contrast, we found that Mtb was incapable of hemolysis over a similar time frame. It is well established that Mm-induced hemolysis is ESX-1 dependent, but our results suggest that there are some differences in Mtb that

affect the rate or extent of hemolysis. One previous study was able to induce hemolysis using Mtb, but this required 48 hr instead of the typical 2 (*Augenstreich et al., 2020*).

## Binding kinetics of ESAT-6 and CFP-10

To measure the binding strength of ESAT-6 and CFP-10, we developed a biolayer interferometry (BLI) assay to measure their interactions (*Figure 3A*). We first measured the ability of ESAT-6 and CFP-10 to form homo-oligomers or hetero-oligomers (*Figure 3B*), and found that they preferentially formed hetero-oligomers. ESAT-6 showed a small, but measurable amount of self-association, while CFP-10 showed no self-association. The dissociation constant ($K_D$) of the ESAT-6/CFP-10 heterodimer has not hitherto been directly measured, but *Renshaw et al., 2002* estimated an upper bound of 10 nM. We performed a detailed measurement of the ESAT-6/CFP-10 association and measured a $K_D$ of 220 pM, indicating exceptionally tight binding (*Figure 3C*). ESAT-6 self-association resulted in reduced overall binding to the sensor and displayed a rapid on rate, but also a rapid off rate (*Figure 3D*). The curve shape is suggestive of something other than traditional 1:1 binding, but for comparison purposes we calculated an apparent $K_D$ of approximately 1.5 µM, weaker than that of the ESAT-6/CFP-10 association.

The pH of the Mtb-containing compartment typically ranges from 4.5 in IFN-γ activated macrophages to 6.2 in non-activated macrophages (*MacMicking et al., 2003*; *Vandal et al., 2008*). Because Mtb can exists in an acidified compartment, we tested whether ESAT-6 binding changes under low pH conditions. Surprisingly, ESAT-6 displayed robust self-association at pH 4.5 (*Figure 3E*), resulting in several-fold more mass attaching to the sensor than even the ESAT-6/CFP-10 association at neutral pH. The apparent $K_D$ for this interaction was slightly weaker than ESAT-6 self-association at neutral pH, but the total amount of binding was 20–30-fold greater, suggesting that low pH conditions trigger ESAT-6 to assemble into larger complexes. Further, there have been conflicting reports on whether ESAT-6 and CFP-10 dissociate from each other at low pH (*de Jonge et al., 2007*; *De Leon et al., 2012*; *Lightbody et al., 2008*). Our results support the stable interaction of ESAT-6 and CFP-10 at low-pH (*Figure 3F*), which exhibits an extremely tight apparent $K_D$ of 0.4 pM at pH 4.5: however, this is likely due to the multivalent ESAT-6 self-association occurring in this condition (*Figure 3G*). To determine the precise pH at which ESAT-6 self-association becomes dominant, we performed BLI experiments at half pH units from 4.5 to 6.5 (*Figure 3H*). We found that 5.0 was the most basic pH at which increased self-association occurred.

## Formation of large ESAT-6 complexes at pH 4.5

The response from pH 4.5 ESAT-6 biosensors appeared to increase past the typical association time of our BLI assay. Because of this, we extended the binding time to 20 min (*Figure 4A*), but the increasing self-association of ESAT-6 continued at roughly the same rate throughout. To explore the possibility of complex formation and to determine the timeframe over which this occurs, we next performed turbidity assays in which 50 µM of either ESAT-6 or CFP-10 were brought to pH 4.5 or 7.5 and $A_{350}$ was measured by plate reader for 1 hr (*Figure 4B*). We observed an increase in turbidity at pH 4.5 but not 7.5 which was greater for ESAT-6 than CFP-10, in line with previous findings (*De Leon et al., 2012*).

The BLI experiments we have shown thus far using the streptavidin-coated biosensors (*Figure 3A*), have utilized ESAT-6 which had been biotinylated using succinimide chemistry (ChromaLINK, Vector labs) allowing for precise quantification of incorporated biotin. Using three molar equivalents of labeling reagent, we obtained ESAT-6 with an average of 0.99 biotins/protein. However, succinimide chemistry can label any primary amine including the N-terminus and lysine residues. ESAT-6 normally contains three lysines, and the pMRLB.7 construct contains one additional lysine for a total of four (*Figure 4C*). Because it is unknown which region of the protein may be involved in self-association, we generated an additional ESAT-6 construct capable of C-terminal site-directed biotinylation via sortase-mediated transpeptidation via an LPETG motif (*Figure 4C*; *Guimaraes et al., 2013*; *Popp et al., 2009*). We performed additional BLI experiments using the ESAT-6-LPETG construct and were able to verify similar behavior at neutral and acidic pH (*Figure 4D–F*).

## Stoichiometry of ESAT-6 complexes

We next visualized the large ESAT-6 complexes by fluorescence microscopy in order to get an idea of their scale. We mixed unmodified ESAT-6 with ESAT-6-LPETG-Biotin at a 10:1 ratio and incubated

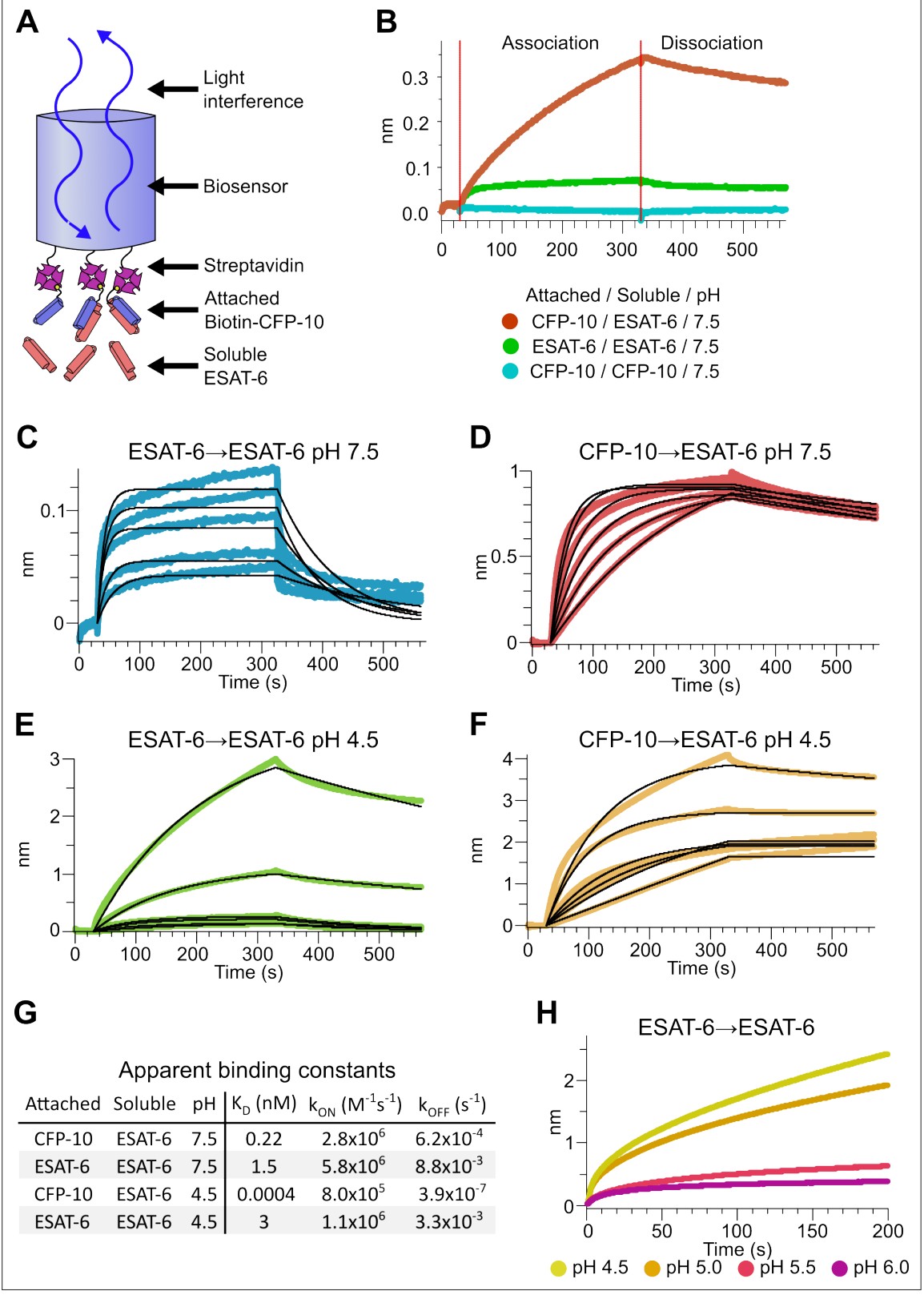

**Figure 3.** Binding kinetics of ESAT-6 and CFP-10 under neutral and acidic conditions. (**A**) Biolayer interferometry (BLI) experimental design. Biotinylated ESAT-6 or CFP-10 are attached to streptavidin coated biosensors, then dipped into solutions of free ESAT-6 or CFP-10. (**B**) BLI experiment depicting ESAT-6 and CFP-10 association, ESAT-6 self-association, and CFP-10 self-association at pH 7.5 with 50 µM. BLI experiment showing (**C**) ESAT-6 self-association at pH 7.5, (**D**) CFP-10/ESAT-6 association at pH 7.5, (**E**) ESAT-6 self-association at pH 4.5, and (**F**) CFP-10/ESAT-6 association at pH 4.5

*Figure 3 continued*

Proteins in (**C–F**) were tested at 1, 3.16, 10, 31.6, and 100 μM. (**G**) Table of apparent binding constants calculated with a 1:1 model for each indicated condition. (**H**) BLI experiment testing ESAT-6 self-association at pH 4.5, 5.0, 5.5, and 6.0 with 100 μM protein.

on a poly-lysine coated coverslip, fixed, then stained with streptavidin-AF488 to visualize ESAT-6 complexes. Fluorescence microscopy shows the development of individual strands with the appearance of a heavily kinked thread or beads on a string (*Figure 5A*). These structures were not present at neutral pH or when 6 M guanidine was added to acidified ESAT-6.

To analyze the stoichiometry of these complexes, we next performed SEC coupled with multi-angle light scattering (MALS), which can measure the molecular weight of complexes as they elute from the SEC column. At pH 7.5, we observed a single peak with an approximate molecular weight of 21 kDa, which most likely corresponds to an ESAT-6 homodimer (*Figure 5B*). We did not observe the presence of any monomeric ESAT-6, indicating that it likely self-pairs in the absence of CFP-10. This agrees with our BLI results from (*Figure 3C*). At pH 4.5, we were surprised to see a single, well-defined peak at 46 kDa, most likely corresponding to a tetramer (*Figure 5C*). No measurable monomer, dimer, or higher molecular weight complexes were observed. While it is not surprising that the high molecular weight complexes observed (*Figure 5A*) were unable to traverse the SEC column intact, it was unexpected that tetramers would be the only species observed.

To address the question of the likely conformation of the homodimer and tetramer, we performed modeling experiments using docking followed by molecular dynamics (MD) simulations to estimate the most stable conformations. We started with a homology model based on previously reported structures of the ESAT-6/CFP-10 heterodimer (PDB: 3FAV, 4J11, 4J7K, 4J10, and 4J7J; *Fan et al., 2015*; *Poulsen et al., 2014*). To estimate the structure of the ESAT-6 monomer, we then performed a molecular dynamics simulation to observe the structural stability of the protein's conformation. We observed that the energy-minimized structure of monomeric ESAT-6 strongly resembles its conformation in the heterodimer. The helix-turn-helix structure exhibits strong amphipathic nature, with both helices aligning to form a hydrophobic face flanked by charged residues (*Figure 5D*).

We next modeled an ESAT-6 homodimer, starting again with a homology model based on the heterodimer followed by a molecular dynamics simulation. The homodimer model showed head-to-tail alignment of the ESAT-6 monomers associating via their hydrophobic faces and stabilized on either side by salt bridges (*Figure 5E*). This homodimer model represents a likely conformation of the species observed in the pH 7.5 SEC-MALS experiment (*Figure 5B*).

To model the tetrameric species observed in the acidic SEC-MALS experiment (*Figure 5C*), we performed docking of ESAT-6 homodimers and then a molecular dynamics simulation of the lowest energy conformation observed in docking. We observed the dimers interacting in a head-to-tail orientation via their hydrophilic faces, supported by salt bridges (*Figure 5F*). In the tetramer model, some of the charged residues, such as $Lys_{67}$, which had previously supported the dimer interface, switched orientations to support the tetramer interface.

For each of these models, the molecular dynamics informs us about the proteins' conformational stability. The root mean square fluctuation (RMSF) quantifies the amount of movement along the peptide backbone for each residue, with lower values equating to more stability. We performed molecular dynamics experiments for ESAT-6 in monomer, dimer, and tetramer forms in order to evaluate the stability of each species (*Figure 5G*). In general, the α-helices of ESAT-6 are stable, while the C- and N-termini are highly flexible. However, in the monomeric form, the residues between 20 and 50 exhibit increased flexibility at pH 7.5. At pH 4.5, the monomeric form exhibits increased flexibility along a majority of the backbone. At both pH values, dimerization (and tetramerization) result in greater overall stability. At pH 7.5, the average RMSF of the monomer is 2.51, which reduces to 2.33 for the dimer due to the reduced flexibility between residues 20 and 50. The tetramer is only marginally more stable, at 2.27. At pH 4.5, the difference between each state is even more pronounced, with an RMSF of 2.74, 2.04, and 1.94 for the monomer, dimer, and tetramer, respectively.

## Generation and characterization of an ESAT-6 specific nanobody

We generated an ESAT-6-specific alpaca-derived nanobody using our previously reported phage display method (*Alfadhli et al., 2021*; *Bachran et al., 2017*; *Weinstein et al., 2022*). E11rv was generated by immunization and panning against pMRLB.7 ESAT-6. Following initial isolation, we performed

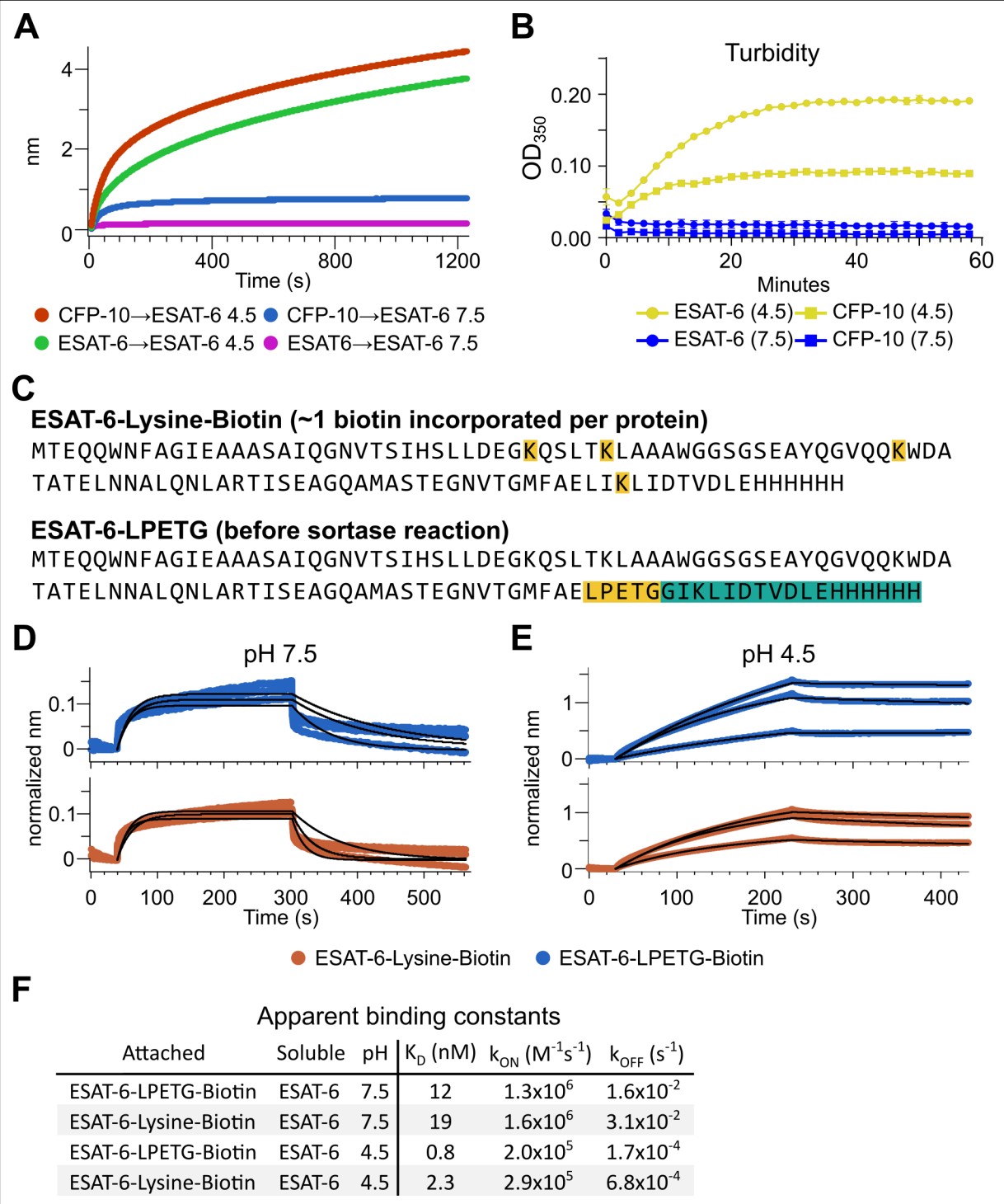

**Figure 4.** Extensive self-association of ESAT-6. (**A**) BLI long-association experiment showing 50 µM ESAT-6 at pH 4.5 or 7.5 attaching to biosensors coated with either ESAT-6 or CFP-10. (**B**) Turbidity assay showing 50 µM ESAT-6 or 50 µM CFP-10 at pH 4.5 or 7.5 over time. Absorbance at 350 nm read every 2 min. (**C**) Lysine-labeled Biotinylated ESAT-6 has an average of 1 biotin per protein attached to the N-terminus or one of the orange highlighted lysine residues. The ESAT-6-LPETG construct amino acid sequence is shown which includes and inserted LPETG motif, in orange. Reaction with biotinylated poly-glycine via sortase-mediated transpeptidation results in the teal highlighted residues being removed and replaced with a biotin molecule. BLI experiments at (**D**) pH 7.5 and (**E**) pH 4.5 showing unmodified soluble ESAT-6 interacting with streptavidin biosensor tips loaded with ESAT-6-Lysine-Biotin (red) or ESAT-6-LPETG-Biotin (blue). Curves were normalized by dividing response values by the amount of protein added during the loading step. (**F**) Summary of apparent kinetic binding constants in (**D–E**).

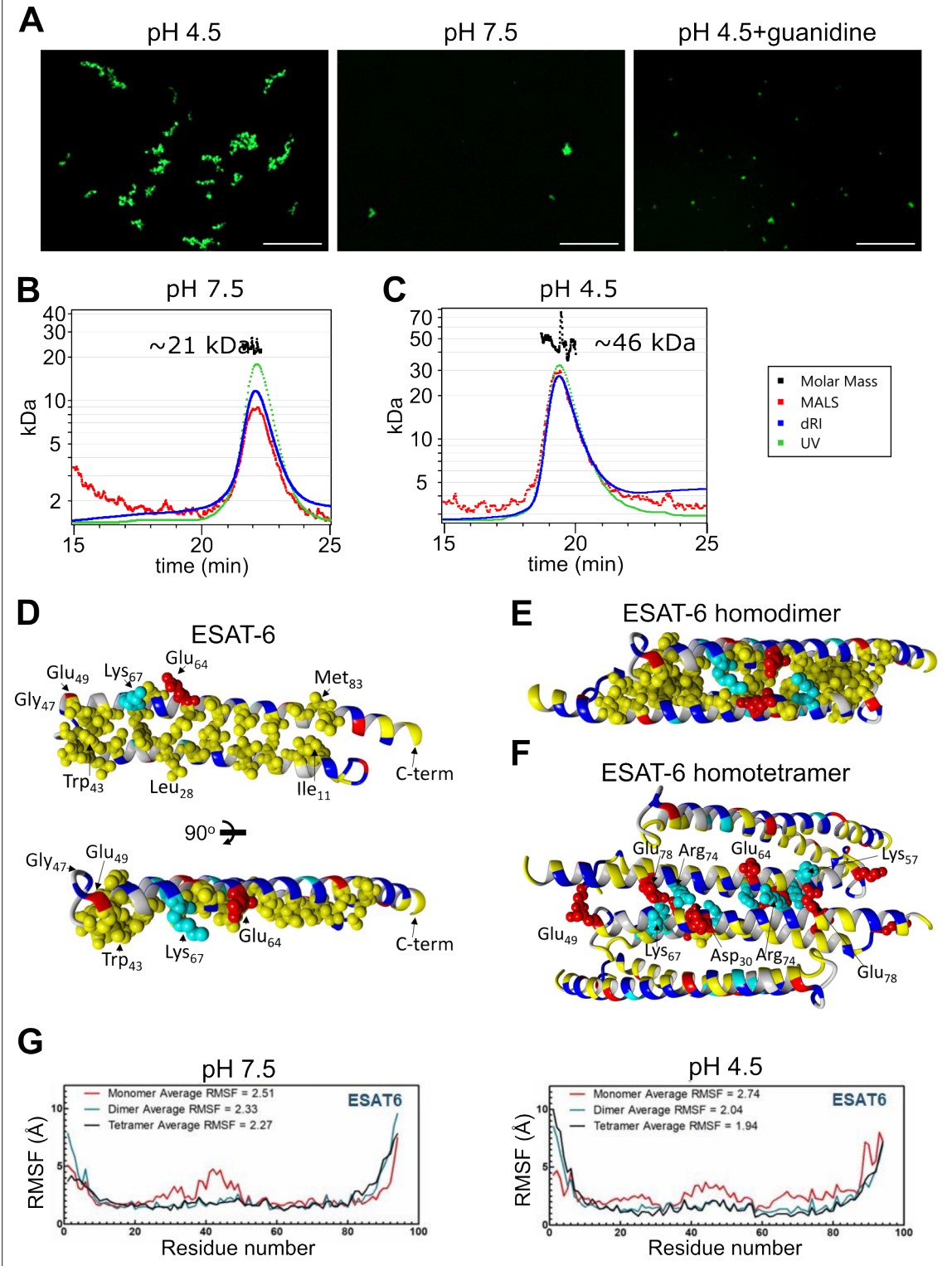

**Figure 5.** Stoichiometry of ESAT-6 self-interaction. (**A**) Fluorescence microscopy showing fluorescently labeled ESAT-6 complexes under different conditions: pH 4.5, pH 7.5, and pH 4.5+6 M guanidine. The images shown are representative of three replicates. Size exclusion chromatography followed by multi-angle light scattering (SEC-MALS) traces showing UV absorbance, refractive index (dRI), and MALS data along with calculated molar mass over the peak at pH 7.5 (**B**), and pH 4.5 (**C**). Data in (**B–C**) is representative of three replicates. (**D–F**) Modeling of ESAT-6 in monomeric (**D**), homodimeric (**E**), and homotetrameric form (**F**). Hydrophobic residues are in yellow, polar residues are in blue, basic residues are in cyan, and acidic residues are in red. Key side chain residues are shown as spheres. (**G**) Root mean square fluctuation (RMSF) values are shown per residue for ESAT-6 monomer, dimer, and trimer at pH 7.5 and 4.5.

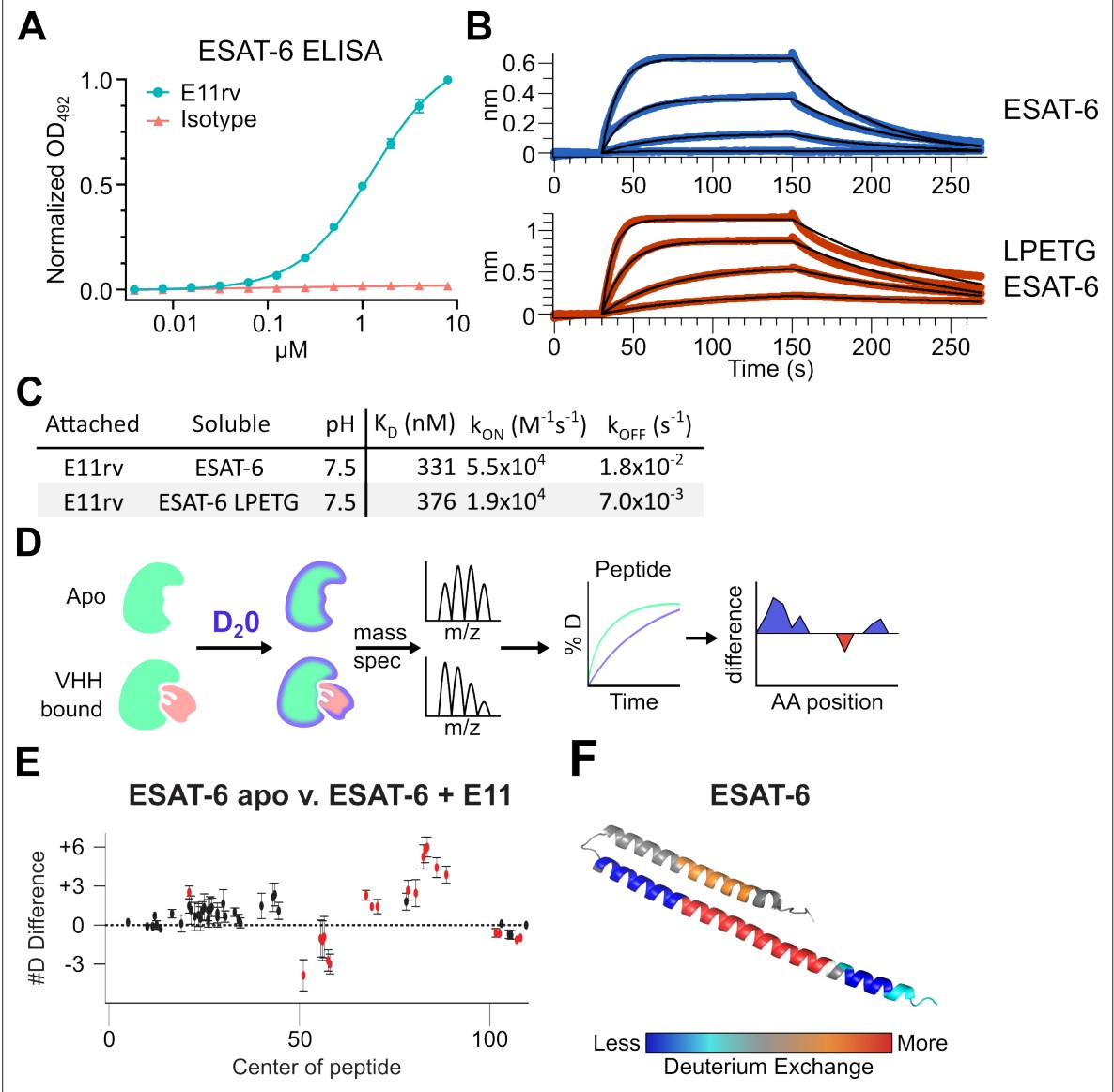

**Figure 6.** An ESAT-6-binding alpaca nanobody. (**A**) ELISA experiment on ESAT-6-coated plates showing nanobody E11rv and an irrelevant isotype control. E11rv displays an EC50 of 17.5 (95% CI 16.0–19.4) µg/mL. (**B**) BLI experiments with E11rv-coated biosensors tested against pMRLB.7 ESAT-6 (blue) or ESAT-6-LPETG (red) at 10, 3.3, 1, and 0.3 µM. (**C**) Summary of kinetic binding constants from (**B**). (**D**) Schematic diagram of hydrogen-deuterium exchange (HDX) experiment showing apo ESAT-6 or E11rv-bound ESAT-6 incubated in D20, then subjected to digestion and mass spectrometry to measure the levels of deuterium incorporation at different times for different peptides, then mapping this to the protein sequence to identify areas with less or greater deuterium exchange between the apo and bound conditions. (**E**) HDX data for ESAT-6 and E11rv showing reduced deuterium uptake around residue 50 and increased uptake around residue 75. (**F**) HDX shifts mapped to the ESAT-6 structure (PDB: 3FAV).

biochemical characterization, starting with ELISA experiments to determine the $EC_{50}$ of E11rv, which was found to be 1.26 µM (*Figure 6A*). We then determined the binding kinetics via BLI, which we performed on both pMRLB.7 ESAT6 as well as our ESAT-6-LPETG construct to ensure that the tag was not essential for binding (*Figure 6B–C*). E11rv bound similarly to ESAT-6 and ESAT-6-LPETG with $K_{D}$s of 331 nM and 376 nM, respectively.

To determine the binding site, we performed hydrogen deuterium exchange experiments with ESAT-6 (*Harris et al., 2023*; *Masson et al., 2019*; *Rathinaswamy et al., 2021*). This involves incubating either apo or E11rv-bound ESAT-6 in $D_2O$ buffer for different lengths of time. The deuterium will exchange with acidic protons in the protein at different rates depending on the solvent exposure, giving insights into which areas of the protein are more tightly folded, or shielded by protein-protein

interactions with a nanobody (*Figure 6D–F*). Digestion followed by MS allows quantification of the extent of deuteration along the peptide backbone and concomitant binding site estimation. For E11rv, we observed decreased deuteration compared to apo around positions 50–65 (on the C-terminal helix, near the loop), and to a lesser extent positions 98–110 (within the tag). The region between these segments, from residues 66 to 94, displayed an increase in solvent exposure, as did residues 14–28, on the opposing helix directly across from 66 to 94. A likely explanation for this pattern is that E11rv binds around residues 50–65, and that this somehow destabilizes the C-terminal portion of the helix or separates the helices enough to allow more solvent exposure between helices. It is unclear how the small region within the tag might be involved, but *Figure 6B* suggests that the positioning of the tag has little impact on binding, and this region is predicted to be largely unstructured.

## E11rv inhibits Mtb replication in macrophages

We next performed functional testing of E11rv using two complementary approaches for measuring the growth and viability of internalized Mtb in macrophages. The first of these experiments used a live/dead reporter strain of Mtb based on H37Rv which constitutively expresses mCherry and dox inducible expression of GFP, which has been shown to correlate with CFU count (*Martin et al., 2012*). We infected THP-1 cells with live/dead reporter Mtb pre-incubated with E11rv or an isotype control, then allowed the infection to proceed for 3 days before inducing GFP with doxycycline for 24 hr. After fixing and staining for actin to reveal the location of cells, we performed fluorescence microscopy on the infected cells. We then used CellProfiler to identify all Mtb within images by their red fluorescence and calculated the GFP/RFP ratio for each identified bacterium (*Stirling et al., 2021*). In this experiment, we found that E11rv led to a statistically significant reduction in Mtb viability (*Figure 7A and B*).

The second functional assay we performed was based on continuous growth measurement of luminescent LuxABCDE-expressing Mtb (*Andreu et al., 2010*; *Leddy et al., 2023*). We prepared stable THP-1 cell lines with cytoplasmic expression of E11rv or an isotype nanobody. We then infected the nanobody-expressing THP-1 cells with luminescent Mtb and monitored their growth continuously via plate reader for 5 days, and we found that E11rv reduced the growth rate substantially (*Figure 7C*). Over the full 5 days, the isotype-expressing cells allowed the Mtb to grow to sixfold over its starting value while the E11rv-expressing cells reduced this to just under threefold.

## Discussion

This study has several key findings. Firstly, ESAT-6 undergoes rapid self-association into large complexes at pH levels below 5.0, and this may be preceded by formation of stable tetramers. Second, in the absence of CFP-10, ESAT-6 exists as a homodimer at neutral pH. Third, our nanobody, E11rv, binds to ESAT-6 and can inhibit the growth of Mtb inside cells when treated externally or via cytoplasmic expression.

The issue of incomplete information about the amino acid sequence of one of the most widely used ESAT-6 constructs is surprising, but none of the data we have generated suggested a practical difference with differently tagged constructs. Previous studies have found major differences in function between recombinant ESAT-6 in general when compared with 'native' ESAT-6 purified from Mtb cultures, which are typically purified using much harsher techniques than recombinant protein due to the lack of affinity tags. One attractive theory for the observed difference between recombinant and native ESAT-6 which is gaining traction is N-terminal acetylation being required for ESAT-6 function, but this has not yet been proven in Mtb infection (*Aguilera et al., 2020*; *Collars et al., 2023*; *Mba Medie et al., 2014*; *Okkels et al., 2004*). The most commonly given reason for the importance of N-acetylation is that acetylated ESAT-6 is more easily released from CFP-10 under acidic conditions. Our results suggest that purified ESAT-6 exists in a dimerized state in the absence of CFP-10, but our Native-PAGE (*Figure 2A*) and SEC (*Figure 1C*) demonstrate that purified ESAT-6 has no trouble interacting with CFP-10 to form a 1:1 complex. It is unclear whether the observed 1:1 complex is an ESAT-6/CFP-10 heterodimer formed by exchange between ESAT-6 and CFP-10 homodimers, or if we are instead observing formation of a tetramer composed of one each ESAT-6 and CFP-10 homodimers. Further, our low pH BLI results (*Figure 3F*) suggest that self-associated ESAT-6 has no difficulty interacting with CFP-10.

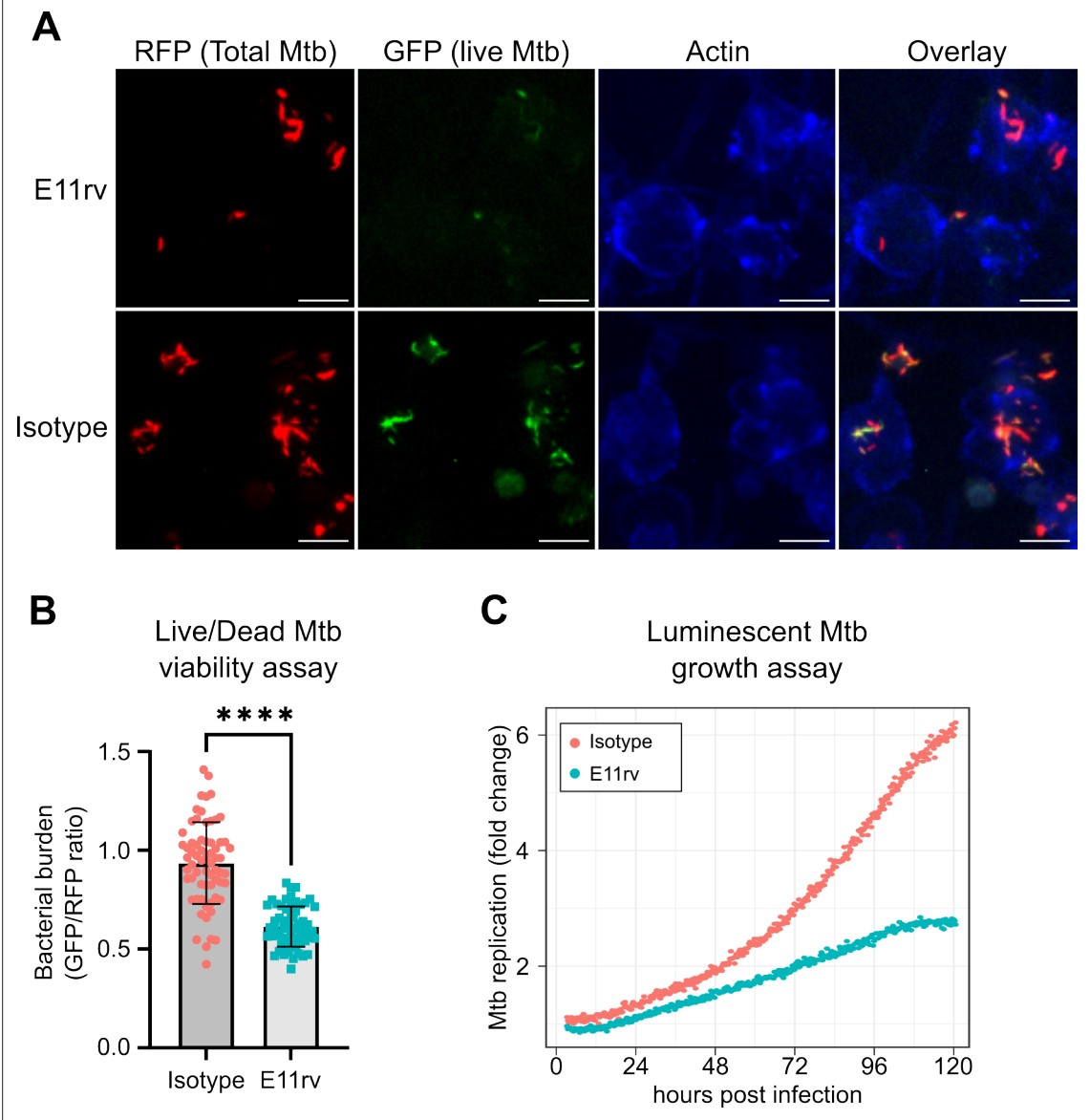

**Figure 7.** E11rv inhibits Mtb growth in cells. (**A**) Representative fluorescence microscopy images for E11rv and isotype-treated cultures. Results include data from three replicates. (**B**) Cellular assay testing the effect of infecting THP-1 cells with Live/Dead Mtb in the presence of E11rv for 3 days, treated with doxycycline for 24 hr to induce GFP expression by remaining metabolically active Mtb. The GFP/RFP ratio was calculated for each Mtb spot identified by fluorescence microscopy. Error bars represent the 95% confidence interval. Statistical significance was determined with a two-tailed unpaired t test. *=0.05, **=0.01, ***=0.001, ****=0.0001. (**C**) Luminescence growth assay of stably transfected THP-1 cells expressing E11rv or isotype nanobody in their cytoplasm. Cells were infected with luminescent Mtb and monitored by luminescence plate reader for 120 hr. Results are representative of 6 replicates.

Our modeling and SEC-MALS data suggest that ESAT-6 can interact with itself using both of its faces, one of which is hydrophobic, and one of which is hydrophilic; and both of which are supported by salt bridges formed by charged residues at either edge. This implies that ESAT-6 may still be able to self-associate within membranes via its hydrophilic face. This agrees with the putative requirement for compartment acidification in order to drive membrane damage, as our data suggests that low pH is required for self-association at this interface, but likely does not affect the hydrophobic interface, which must then require a different mechanism for dissociation and membrane insertion. Some have suggested that other mycobacterial factors may help facilitate the membrane insertion of ESAT-6, such as phthiocerol dimycocerosates (*Augenstreich et al., 2020*; *Augenstreich et al., 2017*).

Modeling approaches have also been used previously to propose an organized ESAT-6 pore structure using similar logic, but these have yet to be supported by physical evidence (*Karki et al., 2020*; *Refai et al., 2015*). However, there is precedent for ESAT-6-like WXG proteins to form organized membrane channels, with a recent study demonstrating a pentameric EsxE/EsxF structure capable of membrane insertion and toxin trafficking (*Tak et al., 2021*). Critically, EsxE does not dissociate from EsxF in order to insert into the membrane. Such structures have yet to be observed with ESAT-6/CFP-10, but this may just be a matter of finding the appropriate conditions and reagents.

Finding molecular tools to interrogate the function of Mtb virulence factors remains a challenge due to the complexity of accessing Mtb in its native environment within the phagosome. The nanobody described here presents a novel way of targeting the bacteria in situ, which we demonstrated by modulating its growth using intracellularly expressed E11rv. Because E11rv showed functional effects both when added in the media and when expressed inside the cell cytoplasm, it suggests a significant amount of communication between Mtb inside the phagosome and other compartments of the cell. This is consistent with the findings of many other studies which show numerous putative activities for ESAT-6 such as induction of apoptosis (*Behura et al., 2019*; *Choi et al., 2010*; *Lim et al., 2016*; *Yang et al., 2015*), inhibition of interferon-γ production (*Kumar et al., 2012*; *Peng et al., 2011*), inhibition of autophagy (*Dong et al., 2016*; *Romagnoli et al., 2012*; *Yabaji et al., 2020*), inhibition of antigen presentation (*Sreejit et al., 2014*), induction of type I interferon (*Jang et al., 2018*; *Lienard et al., 2020*), and a more recent study showing an ESX-1-dependent mechanism for incorporation of Mtb antigens into MHC-I presentation via cytosolic antigen processing (*Leddy et al., 2023*). Most of these functions rely on ESAT-6 getting into the cytoplasm, and this may occur via disruptions in the phagosomal membrane to establish two-way communication between compartments, or by complete phagosomal escape of Mtb into the cytoplasm and subsequent secretion of ESAT-6 directly into the cytoplasm.

## Methods
### Plasmids and primers
Plasmids

| Name | Description | Source |
|---|---|---|
| pMRLB.7 | ESAT-6 expression vector | BEI Resources, NR-50170 |
| pMRLB.46 | CFP-10 expression vector | BEI Resources, NR-13297 |
| pMRLB.7-LPETG | ESAT-6-LPETG expression vector | This study |
| pHEN | Nanobody expression vector | *Hoogenboom et al., 1991* |
| pInducer20 | Lentiviral expression vector | *Meerbrey et al., 2011* |

Primers

| Name | Sequence |
|---|---|
| T7 (sequencing primer) | TAATACGACTCACTATAGGG |
| ESAT-6-Fw | CCGGAAACCGGCGGCATCAAGCTTATCGATACCGTCGACCTC |
| ESAT-6-Rv | GCCGCCGGTTTCCGGGAGCTCTGCGAACATCCCAGTG |

### Cloning ESAT-6-LPETG
ESAT-6-LPETG was generated by in vivo assembly (IVA) as described previously (*García-Nafría et al., 2016*). PCR was performed on pMRLB.7 using ESAT-6-Fw and ESAT-6-Rv primers using Q5 polymerase with the following recipe and cycle design. Reactions were treated with DpnI for 15 min at 37 °C. 2 μL of reaction mixture was added directly to 50 μL of chemically competent DH5-α cells, incubated on ice for 30 min, heat shocked at 42 for 30 s, then immediately added to 200 μL of SOC, after which it

was recovered for 30 min at 37 °C and plated on Luria Bertani (LB)-ampicillin (amp) agar plates, and incubated overnight at 37 °C. Clones were sequence confirmed before further use.

## Expression and purification of ESAT-6 and CFP-10

pMRLB.7, pMRLB.46, and pMRLB.7-LPETG were transformed into *Escherichia coli* BL21 Rosetta rare codon competent cells. Twenty-five mL Luria Bertani (LB) starter cultures were grown overnight at 37 °C with shaking with 100 µg/mL ampicillin to stationary phase. Each starter culture was used to inoculate 1 L of LB, which was then grown at 37 °C until reaching an $OD_{600}$ of 0.8, at which point it was induced with 0.1 mM IPTG for 4 hr at 37 °C. Bacteria was pelleted by centrifugation at 6000 × *g* for 10 min, then frozen at –80 °C. Pellets were thawed and resuspended in 25 mL of 10 mM Tris HCl, 300 mM NaCl, 10% glycerol, 10 mM Imidazole, 1 µg/mL DNase and RNase, 0.5 mg/ml Lysozyme, 10 mM DTT, and 1 tablet/100 mL Sigma fast EDTA free protease inhibitor (Lysis buffer) for 20 min. Lysates were sonicated on ice for 30 s on and 30 s off for a total of 5 min. Crude lysates were then clarified by centrifuged at 20,000 × *g* for 20 min. Supernatant was decanted and spun again similarly. Ni-NTA beads were washed with 10 mM Tris HCl, 300 mM NaCl, 10% glycerol (wash buffer). Lysate was applied to 0.5 mL of Ni-NTA beads per liter of bacteria and rocked for 1 hr at 4 °C then poured over a 25 mL chromatography column. The column was then washed with 10 column volumes of wash buffer plus 10 mM Imidazole, and then eluted in 10 mL wash buffer plus 250 mM Imidazole. Protein containing fractions (assessed by SDS-PAGE) were pooled and imidazole was then buffer exchanged and concentrated with 3 kDa MWCO centrifugal concentrator. Protein concentration was assessed by $A_{280}$, correction for extinction coefficient. Purity was determined by SDS-PAGE, and $\alpha$-$His_6$ western blot.

## Western blot

15% SDS-PAGE gels were run with 1 µg of protein, and transferred to a PVDF membrane (0.2 µm pore size) for 1 hr at 4 °C, using a wet transfer tank (Trans-blot, BioRad). The membrane was rinsed with deionized (DI) water and blocked with 2% non-fat dry milk (Lab Scientific #978-907-4243) in tris buffered saline with 0.1% Tween-20 (TBST) for 30 mins at room temperature (RT). Blots were stained with 1:10,000 $\alpha$-$His_6$-HRP (Invitrogen, MA1-80218) in blocking buffer overnight at 4 °C. Blots were then washed with PBST three times for 5 min each. Signal was developed using SuperSignal Western Pico PLUS (Thermo Fisher Scientific) chemiluminescent substrate according to the manufacturer's instructions and visualized on an ImageQuant LAS 4000.

## Size exclusion chromatography (SEC)

SEC was performed on a HiLoad 16/600 Superdex 75 pg (Cytiva) column on an ÄKTApurifier FPLC (GE) system. Sterile filtered and degassed 10 mM Tris HCl, 300 mM NaCl, 10% glycerol, pH 7.8 was used as a mobile phase and was run at 0.8 mL/min at 4 °C. Data was collected with Unicorn 5.31 (GE).

## Native polyacrylamide gel electrophoresis (PAGE)

15% polyacrylamide gels were cast with 3.5 mL 30% acrylamide:0.8% bis-acrylamide, 1.7 mL of 1.5 M Tris HCl buffer, 70 µL of 10% ammonium persulfate, and 7 µL of TEMED per gel. To analyze the dimerization of CFP10 and ESAT6, 25 µM stocks of each protein were made, mixed in varying ratios, and incubated for 60 min at 4 C. Non-denaturing loading buffer was added and reactions were loaded then run at 4 °C overnight at 50 V in Tris-Glycine buffer without SDS (National Diagnostics) and then Coomassie stained.

## Hemolysis
### Bacterial hemolysis

Hemolysis experiments were performed as described previously (*Conrad et al., 2017*). Sheep blood (HemoStat labs, SBC100) was washed with PBS by centrifugation at 3200 × *g* for 5 min and resuspended to a final concentration of 1% (v/v) red blood cells (RBC) in PBS. Bacterial cultures, Mm, Mtb, MtbΔRD1 were grown in Middlebrook 7H9 with OADC to an $OD_{600}$ between 0.3 and 1.0. Cultures were washed with PBS by centrifugation at 500×g for 5 min and resuspended in PBS. The equivalent of 3 mL of culture at $OD_{600}$ 1.0 was resuspended in 100 µL of PBS and combined with 100 µL of 1% RBCs. Centrifuged samples at 3200 × *g* for 5 min, then incubated the pelleted samples at 37 °C for

Mtb and MtbΔRD1, or 33 °C for Mm for 2 hours. Samples were then resuspended by pipette and spun again similarly. The absorbance at 405 nm was measured for each sample using a plate reader, as well as for samples without bacteria and samples with 0.1% Triton-X100. Percent hemolysis was calculated as $(A_{405}Sample - A_{405}PBS) \div (A_{405}Triton - A_{405}PBS) \times 100$.

## Protein hemolysis

For measurement of protein hemolysis, RBCs were resuspended to 1% (v/v) in McIlvaine buffer of pH 4.5, 5.5, 6.5, 7.5, and 8.5 prepared as previously described (*McIlvaine, 1921*). A total of 100 μL of 1 mg/mL ESAT-6 was combined with 100 μL of 1% RBC solution and incubated at RT for 1 hr. Samples were then spun at 3200 × *g* for 5 min and the $A_{405}$ supernatants were measured by plate reader. PBS and Triton samples were prepared similarly to bacterial hemolysis, and % hemolysis was calculated using the same formula.

## Biolayer interferometry (BLI)

BLI was performed using an Octet Red384 machine (ForteBio). Octet Streptavidin Biosensors (Sartorius, #18–5019) were used for all experiments. Biotinylated proteins were prepared using the succinimide (ESAT-6, CFP-10, RBD) or sortase (ESAT-6-LPETG, E11rv, VHH 52) method. Recombinant SARS-Cov-2 spike RBD was used as a loading control for ESAT-6 and CFP-10 experiments, while VHH 52 against influenza was used as a loading control for E11rv experiments. For all experiments, Biosensors were soaked for 20 min in DI water, blocked for 30 s in 10 mM Citrate, 100 mM NaCl, 3 mM EDTA, 0.1% BSA, 0.005% Tween-20 (running buffer) which was adjusted to match the pH of the test condition (4.5, 5.0, 5.5, 6.0, or 7.5). Biotinylated protein was then loaded onto the biosensors until approximately 1 nm of binding response was observed. Excess protein was washed off with three cycles of 15 s in 10 mM glycine pH 1.7 then 15 s in running buffer. Washing was followed by a 30 s baseline step in running buffer, an association step with test analyte diluted in running buffer, then dissociation step in plain running buffer. When multiple conditions were tested, the lowest concentration was tested first, and hree wash cycles were performed between conditions. Loading control biosensors followed the same protocol with identical sample concentrations. Data was collected with Octet Data Acquisition 10.0.0.87 (ForteBio), and data was analyzed with Octet Data Analysis HT 10.0.0.48 (ForteBio). Background subtraction was performed using loading control biosensors, data was aligned to the average of the baseline step, and smoothed by Savitzky-Golay filtering. All data were fit to a 1:1 model using the "full (assoc and dissoc) setting" using the entire step length. Reported kinetic binding constants ($K_D$, $K_{ON}$, and $K_{OFF}$) values were calculated by taking the geometric mean of binding constants calculated for individual conditions and replicates. Conditions were excluded from the average when the Data Analysis HT software was unable to calculate all constants for a given condition due to insufficient signal. The only conditions excluded using this criterion were ESAT-6 pH 7.5 self-association at 0.316 nM, and E11rv association with ESAT-6 and ESAT-6-LPETG at 0.33 and 0.1 μM.

## Turbidity assay

ESAT-6 was buffer exchanged into PBS using a Zeba spin column (ThermoFisher) and diluted to a concentration of 50 μM. 30 μL of ESAT-6 was added to each well in a 384 well plate. To each well, 2 μL of either PBS or Citrate buffer was added. Citrate buffer contained 50 mM citric acid and 100 mM NaCl, and was adjusted for pH such that mixing 5 mL buffer with 45 mL PBS resulted in a solution with pH 4.5. Plates were immediately added to a plate reader following pH adjustment and $A_{350}$ was read every 2 min for a total of 60 min with 10 s of shaking between each read. Turbidity was calculated as $A_{350}Sample - A_{350}PBS$.

## Biotinylation
## Succinimide biotinylation

Recombinant ESAT-6, CFP-10, and RBD (lacking an LPETG tag) were biotinylated using the ChromaLINK biotinylation kit (Vector labs). Manufacturer's instructions were used. 3 molar equivalents of biotinylation reagent yielded a labeling level of 0.99 biotins/protein calculated using the E1% method in the manufacturer's labeling calculator. Proteins were buffer exchanged into PBS by passing through a PBS-equilibrated Zeba desalting column (Thermo Fisher, 89882). Protein aliquots were snap frozen in liquid nitrogen and stored at –80 °C until use.

## Sortase biotinylation

ESAT-6-LPETG and LPETG-tagged nanobodies were C-terminally biotinylated using sortase as previously described (*Guimaraes et al., 2013*; *Popp et al., 2009*). Briefly, 20 µL of 20 mg/mL sortase enzyme, 20 µL of 10 mM GGG-Biotin peptide, and 200 µL of 1 mg/mL LPETG-labeled protein. This mixture was incubated at 4 °C overnight, then added to 50 µL of packed, PBS-washed Ni-NTA beads to remove the His-tagged sortase enzyme and any unreacted protein for 2 hr at 4 °C. The resulting supernatant was buffer exchanged into PBS by passing through a PBS-equilibrated Zeba desalting column (Thermo Fisher, 89882). Protein aliquots were snap frozen in liquid nitrogen and stored at –80 °C until use.

## Protein microscopy

Coverslips were coated in poly-L-Lysine at 4 °C overnight, washed once with PBS, then dried before use. A 9:1 molar ratio of ESAT-6:ESAT-6-LPETG-Biotin (final concentration 0.24 mg/mL) was prepared prior to addition of 10×citrate buffer prepared similarly to the turbidity assay. The guanidine-containing sample used a protein concentration of 0.7 mg/mL and included a final concentration of 2 M guanidine. The treated protein mixtures were incubated at RT for 30 min. Five 2 µL drops were added to each coverslip and incubated at RT for 30 min. Coverslips were fixed by addition of 4% formaldehyde in PBS for 30 min at RT, washed three times with PBS, blocked in 2.5% BSA in PBS for 30 min at RT, washed three times with PBS, stained with 1:10,000 (in PBS) streptavidin-AF488 (Jacskon Immuno, 016-540-084), and finally washed three times with PBS. Coverslips were mounted onto microscope slides with Prolong Gold antifade (Thermo Fisher, P10144). Images were captured on a Zeiss LSM 980 with Airyscan2 using a 63×oil objective and processed with Zeiss Blue Airyscan joint deconvolution.

## Size exclusion chromatography multi-angle light scattering (SEC-MALS)

SEC-MALS was performed on an AKTApure FPLC (Cytiva) with a DAWN MALS detector (Wyatt). A total of 100 µg of ESAT-6 was buffer exchanged with a Zeba spin column immediately prior to each experiment and injected onto a Superdex 75 increase 10/300 column at a flow rate of 0.5 mL/min at 4 °C. The mobile phase was 10 mM Citrate, 300 mM NaCl, pH 4.5 or pH 7.5. Buffers were sterile filtered to 0.1 µm and degassed prior to use. Data analysis was performed using ASTRA 8.1.2.1 (Wyatt).

## Modeling

### Homology modeling

The ESAT6 and CFP10 protein sequences were modeled via the Homology Modeling protocol in YASARA version 22.09.24. Sidechain optimization occurred through the FoldX plugin, with an energy minimization using AMBER14 force-field. X-ray structures with specified PDB IDs: 3FAV-A, 4J11-B, 4J7K-B, 4J10-A and 4J7J-B (PDB ID-Chain) served as templates for full-length modeling of both proteins, with alternative models considered if alignments were unclear. For CFP10, about 66% aligned with the template, and specific residues (MAEMKTDAAT and RADEEQQQAL) were predicted using loop modeling. Similarly, around 69% of ESAT6 aligned, with residues (STEGNVTGMF and MTEQQW) modeled as described earlier. After side-chain building and optimization, new model components underwent combined steepest descent and simulated annealing minimization. YASARA then integrated the best model parts to create a high-accuracy hybrid for each protein. Structures, dihedrals, and 1D to 3D packing were inspected for optimal monomer models. Dimer models were based on orientations from the three FAV template, while tetrameric structures were predicted using dimer models as templates via the M-Zdock server.

### Molecular dynamic simulations

The protonation status of amino acid residues and subsequent alteration of side-chain interactions and the domain structure can be influenced by the pH, impacting the stability, dynamics, and/or interactions of CFP10, ESAT6, and their heterodimers. As such, these were examined at both pH 4.5 and 7.4 through molecular dynamics simulations. These simulations, performed under the NPT ensemble with periodic boundary conditions, provided a sampling of the conformational space available to the various oligomeric forms of ESAT6 and CFP10. The domains were subjected to MD simulations for a period of 50 ns and snapshots were taken at 250 ps intervals for later examination. Each simulation

process was run twice with distinct random seeds to confirm the reliability of the simulations. The RMSF of each simulation was plotted as displayed in the Figure.

## Nanobody discovery

Nanobody discovery was performed as previously described (*Weinstein et al., 2022*). Briefly, Alpacas were immunized with recombinant ESAT-6 protein and immune serum was collected. Peripheral blood mononuclear cells (PBMC) were isolated and RNA was extracted with an RNeasy mini kit (Qiagen 74104), which was then converted to cDNA using Superscript III (Invitrogen 18080051). VHH genes were amplified using gene specific primers for long- and shot-hinge heavy-chain-only antibodies (*Maass et al., 2007*). Amplified VHH genes were cloned into a phagemid vector based on pCAN-TAB-5E and transformed into competent TG1 *E. coli*. The bacterial library was infected with helper phage to generate a phage library and purified by PEG precipitation (*Frei and Lai, 2016*). The phage library which was panned against recombinant ESAT-6 to enrich strong binders. Panning consisted of incubating the purified phage library in antigen-coated tubes and then eluting with pH 2.2 glycine. Eluted phage were neutralized with 1 M Tris pH 9.1, and transferred to log-phase ER2738 *E. coli*, to produce the enriched library, which was plated on agar. Individual ER2738 clones were selected and screened by ELISA to quantify affinity. E11rv was selected for further testing based on its performance in the screening ELISA, and was cloned into the pHEN periplasmic expression plasmid.

## Nanobody purification

Nanobodies were purified by periplasmic purification. pHEN plasmid containing each nanobody was transformed into *E. coli* WK6 competent cells. Twenty mL starter cultures were grown overnight at 37°C with shaking in Terrific Broth (2% tryptone, 1% yeast extract, 90 mM phosphate) with 100 µg/mL ampicillin to stationary phase. Starter cultures were used to begin 1 L Terrific Broth cultures to be grown at 37° C. When cultures reached OD600 of approximately 0.6 OD, they were induced with 1 mM IPTG overnight at 30°C. Bacteria was pelleted at 4000 x *g* for 10 min and resuspended in 40 mLs of lysis buffer (200 mM HEPES, 0.65 mM EDTA, 0.5 M sucrose, at cold pH 8), then incubated at 4°C for 1–2 hr. Add 40 mLs of ice cold water and incubate for 2 hr, rotating, to lyse periplasm by osmotic shock. Periplasmic fraction was isolated by centrifuging lysate at 8000 x *g* for 15 min, twice. VHH E11 was purified with Ni-NTA chromatography. Purified E11 was then buffer exchanged and concentrated with 3 kDa cutoff centrifuge filters (Millipore), then aliquoted and frozen for future use.

## Enzyme linked immunosorbent assay (ELISA)

ELISA experiments were performed as described previously (*Weinstein et al., 2022*). MaxiSorp plates (Invitrogen 442404) were coated with ESAT-6 at 5 µg/mL in PBS overnight at 4 °C. Plates were blocked in 2% BSA, 2% Polyvinylpyrrolidone (PVP), 0.1% Tween-20 in PBS (blocking buffer) for 30 min at RT. Dilutions of E11rv or isotype control VHH 52 were made in blocking buffer and added to plates for 1 hr at RT with shaking. Plates were washed with PBST three times for 5 min each and biotinylated anti-VHH antibody (Jackson Immuno 128-065-232) at 1:10,000 in blocking buffer was added for 1 hr at RT with shaking. Plates were washed again and incubated with 1:10,000 streptavidin-HRP (Jackson Immuno 016-030-084) in blocking buffer for 1 hr at RT. Plates were washed again and incubated with 50 µL of ODP (Thermo Fisher 34006), prepared according to the manufacturer's instructions, for 15 min before stopping with 50 µL of 2 N $H_2SO_4$. $A_{492}$ was measured using a CLARIOstar PLUS (BMG) plate reader. Raw data was background subtracted with wells lacking nanobody and normalized by dividing by the highest binding well, then fit to a 4PL sigmoid curve in Prism 10.0.0 (Graphpad) to determine the $EC_{50}$.

## Hydrogen-deuterium exchange (HDX)

### HDX-MS sample preparation

HDX reactions comparing ESAT6 Apo to E11rv-bound were carried out in 30 µl reaction volumes containing 15 pmol of ESAT6. A protein mastermix was created for both reaction conditions (500 nM ESAT6, 1 µM E11rv or equivalent volume of E11rv buffer). The exchange reactions were initiated by the addition of 25 µL of $D_2O$ buffer (20 mM HEPES pH 7.5, 100 mM NaCl) to 5 µL of protein mastermix (final $D_2O$ concentration of 78.6% [v/v]). Reactions proceeded for 0.3 s (3 s on ice), 3 s, 30 s, 300 s, and 3000 s at 20 °C before being quenched with ice cold acidic quench buffer, resulting

in a final concentration of 0.6 M guanidine HCl and 0.9% formic acid post quench. All conditions and timepoints were created and run in independent triplicate. Samples were flash frozen immediately after quenching and stored at –80 °C until injected onto the ultra-performance liquid chromatography (UPLC) system for proteolytic cleavage, peptide separation, and injection onto a QTOF for mass analysis, described below.

## Protein digestion and MS/MS data collection

Protein samples were rapidly thawed and injected onto an integrated fluidics system containing a HDx-3 PAL liquid handling robot and climate-controlled (2 °C) chromatography system (LEAP Technologies), a Dionex Ultimate 3000 UHPLC system, as well as an Impact HD QTOF Mass spectrometer (Bruker). The full details of the automated LC system are described in *Stariha et al., 2021*. The samples were run over one immobilized pepsin column (Waters; Enzymate Protein Pepsin Column, 300 Å, 5 μm, 2.1 mm X 30 mm) at 200 μL/min for 3 min at 2 °C. The resulting peptides were collected and desalted on a C18 trap column (Acquity UPLC BEH C18 1.7 mm column (2.1x5 mm); Waters 186003975). The trap was subsequently eluted in line with an ACQUITY 1.7 μm particle, 100×1 mm$^2$ C18 UPLC column (Waters), using a gradient of 3–35% B (Buffer A 0.1% formic acid; Buffer B 100% acetonitrile) over 11 min immediately followed by a gradient of 35–80% over 5 min. Mass spectrometry experiments acquired over a mass range from 150 to 2200 m/z using an electrospray ionization source operated at a temperature of 200 °C and a spray voltage of 4.5 kV.

## Peptide identification

Peptides were identified from a non-deuterated sample of ESAT6 using data-dependent acquisition following tandem MS/MS experiments (0.5 s precursor scan from 150 to 2000 m/z; 12 0.25 s fragment scans from 150 to 2000 m/z). MS/MS datasets were analysed using PEAKS7 (PEAKS), and peptide identification was carried out by using a false discovery-based approach, with a threshold set to 0.1% using a database of purified proteins and known contaminants. The search parameters were set with a precursor tolerance of 20 ppm, fragment mass error 0.02 Da, charge states from 1 to 8, leading to a selection criterion of peptides that had a –10logP score of 16.2.

## Mass analysis of peptide centroids and measurement of deuterium incorporation

HD-Examiner Software (Sierra Analytics) was used to automatically calculate the level of deuterium incorporation into each peptide. All peptides were manually inspected for correct charge state, correct retention time, appropriate selection of isotopic distribution, etc. Deuteration levels were calculated using the centroid of the experimental isotope clusters. Results are presented as relative levels of deuterium incorporation and the only control for back exchange was the level of deuterium present in the buffer (78.6%). Differences in exchange in a peptide were considered significant if they met all three of the following criteria: ≥4.5% change in exchange, ≥0.45 Da difference in exchange, and a p-value <0.01 using a two tailed student t-test. Samples were only compared within a single experiment and were never compared to experiments completed at a different time with a different final $D_2O$ level. The data analysis statistics for all HDX-MS experiments were performed according to the guidelines of *Masson et al., 2019*. The mass spectrometry proteomics data have been deposited to the ProteomeXchange Consortium via the PRIDE partner repository (*Perez-Riverol et al., 2022*).

## Cell lines

Human THP-1 monocytes (ATCC catalog no. TIB-22) were cultured in RPMI medium with 10% FBS (Seradigm) and 1% penicillin-streptomycin (Gibco) at 37 °C and 5% $CO_2$. Cultures were routinely monitored for mycoplasma contamination (abm). Authenticated THP-1 cells were obtained from ATCC; no additional authentication was performed.

## Imaging-based Mtb viability assay

Live/Dead H37rv Mtb constitutively express mCherry and have Tet[ON] inducible GFP expression. THP-1 cells were seeded at 40,000 cells/well in glass bottom, black-walled 96-well tissue culture plates and treated with 100 μM Phorbol-12-myristate-13-acetate (PMA) for 24 hr prior to infection. Endotoxin

free E11rv and unrelated isotype nanobodies were prepared to 1 mg/mL and sterile filtered prior to use. THP-1 cells were infected at an MOI of 1 with Live/Dead Mtb was pre-incubated for 30 min with 100 µg/mL of nanobody. Infected plates were incubated for 3 days in tissue culture conditions. Doxycyline was added to a final concentration of 1 µg/mL and incubated for 24 hr. Plates were fixed with 5% formaldehyde in PBS for 1 hr at RT in accordance with OHSU biosafety procedures before removing plates from the BSL3. Plates were then washed with PBS and stained with Phalloidin-AF405 at 1:40 from 6.6 µM stock for 1 hr. Wells were imaged with a Keyence (BZX-710) using a 20×S Plan Fluor ELWD lens. 49 images were captured for each well using the same exposure settings for each experiment. Images were analyzed with CellProfiler 4.2.5 (*Stirling et al., 2021*). Images with obvious defects such as large dust particles were excluded from the analysis. Mean GFP / Mean RFP was calculated for each identified TB particle and the average GFP/RFP value was tabulated for each image. Images were plotted as individual points in Prism (Graphpad) as described in the statistics section.

## Continuous luminescence viability assay

THP-1 cells were transduced with lentivirus to express nanobody-GFP fusion constructs for E11rv and VHH 52 under a Tet^ON promoter (pInducer20 plasmid). The lentiviruses were generated as described previously (*Niekamp et al., 2021*). THP-1 cells were transduced and selected with 600 µg/mL of G418 for 3 weeks, then maintained with 300 µg/mL G418 in every other passage. For each experiment, opaque white 96-well plates were seeded with 30,000 cells/well along with a matched clear plate, both including 100 nM PMA. Cells were incubated with PMA for 24 hr before inducing with 1 µg/mL doxycycline for another 24 hr. Cell health and nanobody expression was verified by microscopy of the clear plate. The opaque plate was infected with bioluminescent pLux Mtb, which expresses the full luxABCDE cassette, at an MOI of 1. The plate was sealed with a Beathe-Easy plate seal (Research Products International 248738) and put into a climate-controlled CLARIOstart plate reader for continuous luminescence reading for 120 hr.

## Statistics

Statistical analysis was performed in Prism 10.0.0 (Graphpad). In *Figure 2B*, two-tailed one sample t-tests were performed with a significance cutoff of 0.05 against a hypothetical value of 0. In *Figure 7B and a* two-tailed unpaired t test was performed with a significance cutoff of 0.05. For all plots, *=0.05, **=0.01, ***=0.001, ****=0.0001.

## Materials availability

Materials will be made available upon reasonable request to the corresponding author.

# Acknowledgements

This study was supported by the Bill & Melinda Gates foundation grant OPP1179922 (to FGT), NIH R01AI141549 (to FGT), Silver Family Innovation Fund (to FGT), NHLBI training grant 5T32HL083808 (to TAB), Canadian Institutes of Health Research 168998 (to JEB), and the Michael Smith Foundation for Health Research 17686 (to JEB). BLI data were generated on an Octet Red 384, which is made available and supported by the OHSU Biophysics Shared Resources Core and equipment grant number S10OD023413. Mass spectrometry experiments were performed with support from Dr. Larry L David and the OHSU Proteomics Shared Resource core.

# Additional information

### Competing interests

Jessica R Ingram, John E Burke: Hidde L Ploegh: HLP serves as an advisor to and owns stock in Cerberus Therapeutics. HLP serves as a consultant to Johnson and Johnson, Immatics Therapeutics, Cue Biopharma, Revela Therapeutics, and Tiba Bio. The other authors declare that no competing interests exist.

## Funding

| Funder | Grant reference number | Author |
| --- | --- | --- |
| Bill and Melinda Gates Foundation | OPP1179922 | Fikadu G Tafesse |
| National Institutes of Health | R01AI141549 | Fikadu G Tafesse |
| Silver Family Foundation | | Fikadu G Tafesse |
| National Heart, Lung, and Blood Institute | 5T32HL083808 | Timothy A Bates |
| Canadian Institutes of Health Research | 168998 | John E Burke |
| Michael Smith Health Research BC | 17686 | John E Burke |
| OHSU Biophysics Shared Resources Core and equipment | S10OD023413 | Ujwal Shinde |

The funders had no role in study design, data collection and interpretation, or the decision to submit the work for publication.

## Author contributions

Timothy A Bates, Conceptualization, Data curation, Formal analysis, Supervision, Funding acquisition, Validation, Investigation, Visualization, Methodology, Writing - original draft, Project administration, Writing – review and editing; Mila Trank-Greene, Conceptualization, Data curation, Validation, Investigation, Visualization, Methodology, Writing – review and editing; Xammy Huu Wrynla, Aidan Anastas, Ilaria R Merutka, Shandee D Dixon, Abigail R Groncki, Investigation, Methodology, Writing – review and editing; Sintayehu K Gurmessa, Investigation, Writing – review and editing; Anthony Shumate, Jessica R Ingram, Investigation, Methodology; Matthew AH Parson, Data curation, Software, Formal analysis, Validation, Investigation, Visualization, Methodology, Writing – review and editing; Eric Barklis, Conceptualization, Methodology, Writing – review and editing; John E Burke, Ujwal Shinde, Conceptualization, Resources, Formal analysis, Supervision, Funding acquisition, Project administration; Hidde L Ploegh, Conceptualization, Data curation, Software, Formal analysis, Validation, Investigation, Visualization, Methodology, Writing – review and editing; Fikadu G Tafesse, Conceptualization, Resources, Formal analysis, Supervision, Funding acquisition, Validation, Methodology, Writing - original draft, Project administration, Writing – review and editing

## Author ORCIDs

Timothy A Bates ![ORCID] https://orcid.org/0000-0002-2533-7668
Xammy Huu Wrynla ![ORCID] https://orcid.org/0000-0002-7532-8356
Matthew AH Parson ![ORCID] https://orcid.org/0000-0001-6270-559X
John E Burke ![ORCID] https://orcid.org/0000-0001-7904-9859
Fikadu G Tafesse ![ORCID] https://orcid.org/0000-0002-8575-4164

Reviewer #1 (Public Review): https://doi.org/10.7554/eLife.91930.3.sa1
Reviewer #2 (Public Review): https://doi.org/10.7554/eLife.91930.3.sa2
Reviewer #3 (Public Review): https://doi.org/10.7554/eLife.91930.3.sa3
Author response https://doi.org/10.7554/eLife.91930.3.sa4

# Additional files

## Supplementary files
MDAR checklist

## Data availability
Source data for this manuscript can be found at https://doi.org/10.5281/zenodo.10798945.

The following dataset was generated:

| Author(s) | Year | Dataset title | Dataset URL | Database and Identifier |
|---|---|---|---|---|
| Tafesse et al. | 2024 | ESAT-6 undergoes self-association at phagosomal pH and an ESAT-6 specific nanobody restricts M. tuberculosis growth in macrophages | https://doi.org/10.5281/zenodo.10798945 | Zenodo, 10.5281/zenodo.10798945 |

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
