## [Editor Report · eLife assessment]

This **useful** study investigates two secreted Mycobacterium tuberculosis proteins, ESAT-6 and CFP10, using biochemical assays, including a Biolayer Interferometry assay. **Solid** experimental evidence demonstrates that ESAT-6 forms a tight interaction with CFP10 as a heterodimer at neutral pH and that ESAT-6 also forms a homodimer at acidic pH. Additional, more definitive evidence is required to describe how these proteins disrupt the phagosomal membrane. While improved compared to a previous version, the revised manuscript did not address these concerns adequately.

---

## [Referee Report · Reviewer #1 (Public Review)]

Summary:

The authors sought to establish a biochemical strategy to study ESAT-6 and CFP-10 biochemistry. They established recombinant reagents to study these protein associations in vitro revealing an unexpected relationship at low pH. They next develop much needed reagents to study these proteins in an infection context and reveal that treatment with an ESAT-6 nanobody enhances Mtb control.

Strengths:

The biochemical conclusions are supported by multiple configurations of the experiments. They combine multiple approaches to study a complex problem.

Weaknesses:

It would be valuable to understand if the nanobody is disrupting the formation of the ESAT6-CFP10 complex. It is unclear how the nanobody is functioning to enhance control in the infection context. More detail or speculation in the discussion would have been valuable. Where is the nanobody in the cell during infection?

---

## [Referee Report · Reviewer #2 (Public Review)]

Summary:

Bates TA. et al. studied the biochemical characteristics of ESAT-6, a major virulence factor of Mycobacterium tuberculosis (Mtb), as part of the heterodimer with CFP10, a molecular chaperon of ESAT-6, as in homodimer and in homotetramer using recombinant ESAT-6 and CFP10 expressed in *E. coli* by applying several biochemical assays including Biolayer Interferometry (BLI) assay. The main findings show that ESAT-6 forms a tight interaction with CFP10 as a heterodimer at neutral pH, and ESAT-6 forms homodimer and even tetramer based larger molecular aggregates at acidic pH. Although the discussion of the potential problems associated with the contamination of ESAT-6 preparations with ASB-14 during the LPS removal step is interesting, but this research does not test the potential impact of residual ASB-14 contaminant on the biochemical behavior ESAT-6-CFP10 heterodimer and ESAT-6 homodimer or tetramer and their hemolytic activity in comparison with the ones without ASB-14. The main strength of this study is the generation of ESAT-6 specific nanobodies and demonstration of its anti-tuberculosis efficiency in THP-1 cell line infected with Mtb strains with reporter genes.

Strengths:

Generation and demonstration of the anti-ESAT-6 nanobodies against tuberculosis infection in cell line based Mtb infection model. Probably identifying potential anti-ESAT-6 nanobody interacting amino acid residues of ESAT-6 is critical in understanding their effects on ESAT-6 mediated membrane lytic activity.

Weaknesses:

Although the biochemistry studies provide quantitative data about the interactions of ESAT-6 with its molecular chaperon CFP10 and the interaction of ESAT-6 homodimer and tetramers, the novel information from these studies are minimal.

---

## [Referee Report · Reviewer #3 (Public Review)]

Summary:

This manuscript describes some biochemical experiments on the crucial virulence factor EsxA (ESAT-6) of Mycobacterium tuberculosis. EsxA is secreted via the ESX-1 secretion system. Although this system is recognized to be crucial for virulence the actual mechanisms employed by the ESX-1 substrates are still mostly unknown. The EsxA substrate is attracting most attention as the central player in virulence, especially phagosomal membrane disruption. EsxA is secreted as a dimer together with EsxB. The authors show that EsxA is also able to form homodimers and even tetramers, albeit at very low pH (below 5). Furthermore addition of a nanobody that specifically binds EsxA is blocking intracellular survival, also if the nanobody is produced in the cytosol of the infected macrophages.

Strengths:

Decent biochemical characterization of EsxA and identification of a new and interesting tool to study the function of EsxA (nanobody). Well written.

Weaknesses:

The findings are not critically evaluated using extra experiments or controls.

For instance, tetrameric EsxA in itself is interesting and could reveal how EsxA works. But one would say that this is a starting point to make small point mutations that specifically affect tetramer formation and then evaluate what the effect is on phagosomal membrane lysis. Also one would like to see experiments to indicate whether these structures can be produced under in vitro conditions, especially because it seems that this mainly happens when the pH is lower than 5, which is not normally happening in phagosomes that are loaded with M. tuberculosis.

Also the fact that the addition of the nanobody, either directly to the bacteria or produced in the cytosol of macrophages is interesting, but again the starting point for further experimentation. As a control one would like to se the effect on an Esx-1 secretion mutant. Furthermore, does cytososlic production or direct addition of the nanobody affect phagosomal escape? What happens if an EsxA mutant is produced that does not bind the nanobody?

Finally, it is a bit strange that the authors use a non-native version of esxA that has not only an additional His-tag but also an additional 12 amino acids, which makes the protein in total almost 20% bigger. Of course these additions do not have to alter the characteristics, but they might. On the other hand they easily discard the natural acetylation of EsxA by mycobacteria itself (proven for M. marinum) as not relevant for the function because it might not happen in (the close homologue) M. tuberculosis.

---

## [Author Response]

The following is the authors’ response to the original reviews.

**Reviewer #1 (Recommendations For The Authors):**
Will the nanobody be available to the TB research community?

Yes, we will make E11rv available upon request. Please see our materials availability statement.

**Reviewer #2 (Recommendations For The Authors):**
(1) It would be interesting to test the potential impact of residual ASB-14 contaminant on the biochemical behavior of ESAT-6-CFP10 heterodimer and ESAT-6 homodimer or tetramer and their hemolytic activity in comparison with the ones without ASB-14.

We agree that this is an interesting line of questioning. Based on the study by Refai et al. that we cite in the text, ESAT-6 treated with nonionic detergents ASB-14 or LDAO, but not other common detergents, undergoes a conformational change that increases its cytotoxicity in cell assays, hemolytic activity, and ability to dimerize with CFP-10. What is not known at this point, is how similar the ASB-bound conformation is to anything seen physiologically.

(2) Building on the progress in making anti-ESAT-6 nanobodies and their anti-Mtb effects in the cells, it could have been tested in human or mouse primary macrophages infected with Mtb and a mouse model of Mtb infection for its anti-Mtb efficiency.

We thank the reviewer for this suggestion, and we agree that these would be very informative next steps for determining the therapeutic potential of anti-ESAT-6 nanobodies.

**Reviewer #3 (Recommendations For The Authors):**
Minor comments:Line 133: "It is well established that Mm-induced hemolysis is ESX-1 dependent, but our results suggest that Mtb must lack one or more factors necessary for efficient hemolysis.". I would tone this down a bit, as it is also known that M. tuberculosis escapes much later than M. marinum from the phagosome, which could indicate different kinetics.

We thank the reviewer for their insightful comments. We agree that the kinetics of Mtb and Mm infection are quite different and that this may impact the hemolysis assay. As described by Augenstreich et al. some hemolysis by Mtb is observed at 48 hours, though the method of normalization makes it impossible to determine absolute amount of hemolysis that occurred in their experiment. Our findings just show that the absolute amount of Mtb hemolysis in 2 hours is negligible, setting it apart from Mm. We have edited the wording of this statement in the manuscript to avoid any confusion.

Line 155: "Because Mtb often exists in an acidified compartment". First of all, the reference used here does not discuss anything about Mtb, secondly, papers that do measure the acidification of Mtb-loaded phagosomes indicate that this acidification is very mild (typically to pH 6.2).

We agree that this point should be articulated more precisely. We have added additional clarification that the pH of Mtb-containing compartments in macrophages can fall in a broad range depending on the activation state of the macrophages, and that non-activated macrophages are typically only mildly acidic. We have updated our references to better describe the current state of knowledge on this topic.

Line 339: "Whereas most of these functions rely only on the secretion of ESAT-6 into the cytoplasm, the ability of E11rv to access Mtb suggests that this communication is likely two-way." No, not necessary, there are many processes in which ESX-1 substrates affect the macrophage. This nanobody could affect EsxA functioning only once the bacteria reach the cytoplasm. I think checking phagosomal escape in these cells is therefore crucial.

We agree that phagosomal escape and subsequent direct secretion of ESAT-6 into the cytoplasm is a reasonable alternative hypothesis. We have added this point to our discussion, and we agree that looking directly at phagosomal escape is an important next step.

Figure 7 is not mentioned in the text (mistake for Fig 6).

This has been corrected.